# Recent Advances in Targeted Therapies for Advanced Gastrointestinal Malignancies

**DOI:** 10.3390/cancers12051168

**Published:** 2020-05-06

**Authors:** Jasmine C. Huynh, Erin Schwab, Jingran Ji, Edward Kim, Anjali Joseph, Andrew Hendifar, May Cho, Jun Gong

**Affiliations:** 1Hematology Oncology, University of California, Davis, Sacramento, CA 95817, USA; jhkim@ucdavis.edu (E.K.); ajoseph@ucdavis.edu (A.J.); maycho@ucdavis.edu (M.C.); 2Internal Medicine, University of California, Davis, Sacramento, CA 95817, USA; jinji@ucdavis.edu; 3Hematology Oncology, Cedars-Sinai Medical Center, Los Angeles, CA 90048, USA; andrew.hendifar@cshs.org (A.H.); jun.gong@cshs.org (J.G.)

**Keywords:** gastrointestinal cancer, molecular targets

## Abstract

The treatment of advanced gastrointestinal (GI) cancers has become increasingly molecularly driven. Molecular profiling for HER2 and PD-L1 status is standard for metastatic gastroesophageal (GEJ) cancers to predict benefits from trastuzumab (HER2-targeted therapy) and pembrolizumab (anti-PD-1 therapy), while extended RAS and BRAF testing is standard in metastatic colorectal cancer to predict benefits from epidermal growth factor receptor (EGFR)-targeted therapies. Mismatch repair (MMR) or microsatellite instability (MSI) testing is standard for all advanced GI cancers to predict benefits from pembrolizumab and in metastatic colorectal cancer, nivolumab with or without ipilimumab. Here we review recent seminal trials that have further advanced targeted therapies in these cancers including Poly (adenosine diphosphate–ribose) polymerases (PARP) inhibition in pancreas cancer, BRAF inhibition in colon cancer, and isocitrate dehydrogenase (IDH) and fibroblast growth factor receptor (FGFR) inhibition in biliary tract cancer. Targeted therapies in GI malignancies constitute an integral component of the treatment paradigm in these advanced cancers and have widely established the need for standard molecular profiling to identify candidates.

## 1. Introduction

Molecular studies, particularly next generation sequencing, has become more affordable and efficient in the modern era of oncology. As a result, research and development as well as clinical practice in this field has changed drastically. This is particularly true in gastrointestinal (GI) malignancies, which has become increasingly molecularly driven. As more and more molecular targets and targeted therapeutics are elucidated, it also becomes challenging to maintain a comprehensive understanding of each agent’s role in clinical practice. In this review, we break down these practice-changing developments in advanced gastrointestinal malignancies by therapeutic targets.

## 2. Microsatellite Instability (MSI) High

Mismatch repair proteins (MMR) serve to repair insertions or deletions in microsatellites, which are repetitive DNA units. Known MMR gene products are MLH1, MSH2, MSH6 and PMS2. Dysfunction of this system is known as deficient mismatch repair (dMMR) and leads to the accumulation of mutations in microsatellites, known as microsatellite instability (MSI). Tumors with high levels of MSI are deemed MSI-high (MSI-H) [1,2,3]. dMMR is seen in a variety of cancers, such as gastrointestinal (GI), uterine, ovarian and prostate malignancies. The prevalence of dMMR in GI malignancies according to a comprehensive review by Lorenzi et al. was 13% (95% CI 10–16%) for Stage 1–2 cancers, and 10% (95% CI 7–13%) in Stage 3–4 cancer [4]. Colorectal cancer (CRC) has the highest prevalence of dMMR tumors, ranging from 5% to 20% [5,6]. dMMR tumors are associated with a large number of activated CD8+ cytotoxic T-cells and upregulated checkpoints (i.e., PD-1 and PD-L1), which has led to several trials have exploring use of PD-1 and PD-L1 blockade in GI cancers [5,7,8].

Le et al.’s phase II trial compared response to programmed cell death 1 (PD-1) blockade with pembrolizumab in 41 patients with dMMR colorectal adenocarcinomas (11 patients, the majority of whom had received ≥2 chemotherapy regimens), proficient MMR (pMMR) colorectal adenocarcinomas (21 patients) and dMMR non-colorectal cancers (9 patients with predominantly GI cancers). Patients with dMMR tumors, both CRC and non-CRC, had improved immune-related overall response rate (ORR) compared to those with pMMR tumors. The immune-related ORR was 40% (95% confidence interval (CI) 12–74) in dMMR CRC, 71% (95% CI 29–96) in dMMR non-CRC, and 0% in pMMR CRCs (95% CI 0–20). Disease control rates were 90% (95% CI 55–100) in the dMMR CRC group, 71% (95% CI 29–96) in the dMMR non-CRC group and 11% (95% CI 1–35) in the pMMR CRC group. Patients in the dMMR CRC group had median overall survival (OS) and progression-free survival (PFS) that were not reached. These findings lead to the USA Food and Drug Administration (FDA) approval of pembrolizumab in any solid tumor with MSI-high or dMMR in the chemotherapy refractory setting [7].

The original phase II CheckMate-142 trial demonstrated that combination nivolumab (3mg/kg) and low-dose ipilimumab (1 mg/kg) had a durable clinical benefit with tolerable side effects in patients with MSI-H, chemotherapy-resistant, metastatic CRC [2,3,9]. In the recent update of this trial, 45 patients with dMMR/MSI-H metastatic CRC without any prior therapy received nivolumab and low-dose ipilimumab. Data presented at the ASCO GI symposium in January 2020 showed that at a median follow-up of 19.9 months, investigator-assessed ORR was 64% (95% CI 49–78) with a 9% complete response rate. Median duration of response, PFS and OS were not reached. Treatment was also well-tolerated with 20% grade 3–4 AEs. These encouraging findings were the basis for the FDA approval of nivolumab and low-dose ipilimumab in treatment-refractory dMMR/MSI-H metastatic CRC. Studies are ongoing regarding its efficacy as a first-line therapeutic option for dMMR/MSI-H metastatic CRC.

Following these results in the dMMR metastatic CRC population, Chalabi et al.’s ongoing NICHE study (NCT03026140) investigated the utility of neoadjuvant ipilimumab and nivolumab in patients with early-stage CRC. Recently published results of the 35 evaluable patients with early-stage CRC (21 with dMMR and 20 with pMMR) who received one dose of ipilimumab and two doses of nivolumab prior to surgery were very favorable. The dMMR group had a 100% pathological response rate; 19 patients had major pathological responses (defined as ≤10% residual viable tumor) and 12 patients had pathological complete responses. The pMMR group only had 27% major pathological responses, none of which were pathological complete response [10]. While larger scale studies are still needed, these findings are encouraging in that neoadjuvant ipilimumab and nivolumab may have a role for a defined group of early-stage colon cancer patients.

Similar to CRC, there continues to be mounting evidence that first-line pembrolizumab may be favorable over chemotherapy in metastatic MSI-H gastric cancer populations. The KEYNOTE-062 study is a Phase III randomized trial that compared pembrolizumab, pembrolizumab plus chemotherapy (cisplatin and fluorouracil or capecitabine) or placebo plus chemotherapy as first-line therapy in advanced gastric and gastroesophageal junction (GEJ) adenocarcinoma. Over 760 patients with HER2-negative, combined positive score (CPS) ≥1 tumors were included. Single-agent pembrolizumab was found to be non-inferior to chemotherapy in patients with CPS ≥ 1, with the most pronounced effect in tumors with CPS ≥ 10. Median OS was 17.4 months in the CPS ≥ 10 group versus 10.8 months in the chemotherapy group (HR 0.69, 95% CI 0.49–0.97). In the exploratory analysis which included 50 patients with MSI-H tumors with CPS ≥1, median OS was not reached for both the pembrolizumab only group and pembrolizumab plus chemotherapy groups, and ORR was 64.7% and 57.1%, respectively. Median duration of response was not reached in the pembrolizumab plus chemotherapy group and was 21.2 months in the pembrolizumab only group, compared to 7 months in the chemotherapy only group [11]. While clinical practice will likely continue to favor chemotherapy as the first line choice in microsatellite stable gastric cancer populations, this study provided that first-line pembrolizumab may be favorable in MSI-H high populations. [12].

## 3. PD-1/PD-L1

A comprehensive molecular analysis of 295 gastric adenocarcinomas as part of The Cancer Genome Atlas Project identified programmed cell death 1 ligand 1 (PD-L1) in a subset of gastric adenocarcinomas [13]. The KEYNOTE-059 was a three-cohort Phase II trial that included 259 patients with advanced gastric or GEJ adenocarcinoma who had progressed on two or more prior lines of chemotherapy. Patients were given pembrolizumab monotherapy. There was an 11.6% ORR and patients who had PD-L1 positive tumors had a higher ORR compared to those with PD-L1 negative tumors (15.5% vs. 6.4%, respectively). Pembrolizumab was generally well-tolerated with 17.8% of patients experiencing an immune-mediated adverse event (AE) though only 4.6% of patients had Grade 3–4 events [14]. Other immune checkpoint inhibitors have also been explored. The ATTRACTION-2 study, a randomized, placebo-controlled Phase III trial conducted in Asia, compared nivolumab to placebo in a population of 493 heavily pre-treated patients with unresectable advanced or recurrent gastric or GEJ adenocarcinoma. Compared to the placebo group, patients receiving nivolumab had a higher median OS (5.26 months, 95% CI 4.60–6.37, in the nivolumab group, vs. 4.14 months, 95% CI 3.42–4.86, in the placebo group) and response rate (11% vs. 0%, respectively) [15]. Interestingly, median OS in the nivolumab group was similar in the PD-L1 positive and negative groups (5.22 months, 95% CI: 2.79–9.36 vs. 6.05 months, 95% CI 4.83–8.54, respectively) [15]. As a result, pembrolizumab was approved as a third-line option in metastatic GEJ adenocarcinoma with a PD-L1 CPS ≥1 in the USA, while nivolumab is approved in Japan for metastatic gastric cancer progressing after chemotherapy irrespective of PD-L1.

Certain subgroups of gastric cancer patients are known to have preferential response to anti-PD-1/PD-L1 therapy. Kim et al.’s Phase II single-center study conducted in Korea evaluated the efficacy of single-agent pembrolizumab in 61 patients with metastatic gastric cancer who had received at least two prior lines of therapy. While the ORR was 24.6% (95% CI, 14.7–37.3) with a 19.7% PR rate, most notable was that the entire subset of Epstein–Barr Virus (EBV)(+) patients achieved PR with a median duration of response of 8.5 months. One EBV(+) patient with multiple hepatic metastases was actually able to undergo curative surgery following eight cycles of pembrolizumab [16]. A prior study by Derks et al. found that EBV(+) gastric cancers have robust PD-L1 expression, which potentially explains this response [17]. Overall, these findings suggest that EBV positivity might be a predictive biomarker for the use of PD-1 therapy in gastric cancers. Another subgroup of interest is HER2+ gastric cancer patients for which addition of trastuzumab to chemotherapy is currently the standard of care. It is hypothesized that combination anti-PD-1 and anti-HER2 therapy will increase T-cell activation and subsequent antitumor immune response. The KEYNOTE-811 trial (NCT03615326) built on the promising findings of the mEGA study and is an ongoing Phase III study of HER2+, metastatic or unresectable gastric or GEJ adenocarcinoma evaluating chemotherapy, trastuzumab and pembrolizumab or placebo. Phase II data presented at ASCO GI Symposium 2019 was promising with ORR 87% in the group receiving pembrolizumab and in the induction phase of the trial, where patients received trastuzumab and pembrolizumab without chemotherapy, 52% of patients had a reduction in target lesions after just one dose of the doublet therapy [18,19].

Squamous cell carcinoma (SCC) of the esophagus accounts for 90% of metastatic esophageal cancer in Asia, Africa and France, but only accounts for approximately 40% of cases in the United States [20,21]. In the Phase III KEYNOTE-181 study, 628 patients with advanced or metastatic esophageal carcinoma were randomized to receive either pembrolizumab or investigator’s choice of chemotherapy with either paclitaxel, docetaxel or irinotecan. Sixty-four percent of patients had SCC histology. Pembrolizumab was found to be superior to chemotherapy in patients with CPS ≥ 10, with median OS 9.3 months in the pembrolizumab group versus 6.7 months in the chemotherapy group (HR 0.69, 95% CI 0.52–0.93; *p* = 0.0074). The 12-month OS rate for the pembrolizumab group was 43% versus 20% in the chemotherapy group. In patients with SCC, median PFS for pembrolizumab vs. chemotherapy was 3.2 months vs. 2.3 months, respectively; in patients with adenocarcinoma, median PFS was 2.1 months vs. 3.7 months, respectively. Pembrolizumab was also better tolerated with fewer rates of any-grade AEs compared to chemo (64% vs. 86%, respectively) and grade 3–5 drug-related AEs (18% vs. 41%). Based on these findings, pembrolizumab is now FDA approved as a second-line standard of care therapy for patients with advanced or metastatic esophageal SCC and PD-L1 CPS ≥ 10 [22,23].

## 4. HER2

HER2 is overexpressed/amplified in gastroesophageal and gastric cancers, which makes it an attractive therapeutic target in these malignancies [24]. Trastuzumab is a monoclonal antibody that targets HER2. The ToGA trial, a phase III, randomized-controlled trial that included nearly 600 patients with inoperable, locally advanced, recurrent or metastatic adenocarcinoma of the stomach or gastroesophageal junction (GEJ) found that the combination of trastuzumab and chemotherapy (cisplatin plus 5-fluorouracil (5-FU) or capecitabine) had a survival benefit in HER2 positive metastatic gastric or GEJ adenocarcinoma patients. Median overall survival (OS) in the trastuzumab group was 13.8 months versus 11.1 months in the chemotherapy only group (HR 0.74; 95% CI 0.60–0.91; *p* = 0.0046) and objective response rate (ORR) was 47% vs. 35% (OR 1.70) [25]. These results established trastuzumab and chemotherapy as first-line therapy in patients with HER2 positive metastatic gastric or GEJ adenocarcinoma. New HER2-directed therapy with trastuzumab deruxtecan, a novel antibody-drug conjugate composed of a humanized anti-HER2 antibody, cleavable peptide-based linker and topoisomerase I inhibitor, has received accelerated approval in metastatic breast cancer and has shown preliminary efficacy in gastric cancer. Shitara et al.’s Phase I trial to assess safety and preliminary efficacy of trastuzumab deruxtecan included 44 patients with advanced HER2-positive gastric or GEJ cancer. Nineteen patients (43.2%, 95% CI: 28.3–59.0) had a confirmed objective response. Notable AEs were decreased blood counts (16–30% were ≥ Grade 3), and there were four cases of pneumonitis [26]. The Phase II DESTINY-Gastric-01 trial is ongoing in Asia with over 180 patients, comparing trastuzumab deruxtecan to chemotherapy (monotherapy with paclitaxel or irinotecan) in patients with HER2-expressing unresectable or metastatic gastric or GEJ cancer with progression on ≥2 lines of therapy, including trastuzumab and chemotherapy. Preliminary data show results consistent with the Phase I trial [27,28].

HER2 amplification and/or overexpression is seen in 2–6% of patients with colorectal cancer [29]. Several studies have looked at the role of anti-HER2 therapy in metastatic colorectal cancer (mCRC). The MyPathway study was a Phase IIa multiple basket study involving 230 patients with advanced refractory solid tumors harboring HER2, EGFR, BRAF and Hedgehog pathway alterations. Thirty-seven heavily pretreated patients with mCRC with HER2 amplification/overexpression received trastuzumab plus pertuzumab. ORR was 38% (95% CI 23–55) with a median duration of response of 11 months (95% CI 3 months—not estimable) [30]. The HERACLES trial was a Phase II trial that included patients with KRAS wildtype, HER2-positive (defined as 2+/3+ HER2 score in >50% of cells by immunohistochemistry (IHC) or with a HER2:CEP17 ratio >2 in more than 50% of cells by fluorescent in situ hybridization (FISH)) mCRC who had been refractory to standard of care therapy with EGFR 1/2 inhibitors. Twenty-seven patients were given the combination of trastuzumab and lapatinib. ORR was 30% (95% CI 14–50) with one patient achieving a complete response, and median OS was 46 weeks (95% CI 33–68). The most common AEs were diarrhea, rash and fatigue (78%, 48%, and 48% of patients, respectively). These findings suggest that HER2 positivity is an important driver in mCRC. Further supporting this is the fact that patients with higher HER2 gene copy number in the HERACLES trial experienced a longer PFS (29 weeks, 95% CI 19–43, with gene copy number >9.45 versus 16 weeks, 95% CI 3–17, for patients with gene copy number <9.45) and objective response (0 patients versus 8 patients, respectively) [31].

HER2 is also overexpressed in 9–20% of biliary cancers. In 2015, Javle et al. retrospectively evaluated cases of advanced CCA and gallbladder cancer with HER2 overexpression who received HER2-directed therapy. Eight patients were identified and showed an overall improvement, with three achieving disease stability, four achieving PR and one achieving CR [32]. The aforementioned MyPathway study also included HER2-positive metastatic biliary cancer patients who were given combination pertuzumab and trastuzumab. Preliminary results were presented at ASCO 2017. Eleven patients had either HER2 amplified or HER2 mutated biliary cancer. HER2 amplified patients, eight in total, had the best response to therapy. Of the eight patients, clinical benefit was seen in six patients and ORR was 37.5%. Sample size was small and at the time the trial was still accruing additional patients [33].

## 5. BRAF

BRAF is a member of the Raf kinase family and is integral in regulating cell proliferation through the mitogen-activated protein (MAP) kinase pathway. Mutant BRAF has been involved in the pathogenesis of many cancers but can also be seen in benign conditions.

Between 5% and 15% of CRC have BRAF mutations, typically associated with female sex and right-sided colon cancers [34,35]. BRAF V600E mutations account for 95% of activating BRAF mutations and are associated with poor prognosis [36]. It is already well-established in the literature that tumors with extended RAS mutations, including NRAS and non-exon 2 KRAS mutations, BRAF mutations and right-sided colon tumors do not benefit from EGFR inhibition [37,38]. Attempts at monotherapy BRAF inhibition as well as combination BRAF and MEK inhibition, as used in melanoma, have been unsuccessful due to EGFR-mediated adaptive feedback [36,39,40]. However, the BEACON trial, a Phase III trial involving 665 patients with BRAF V600E-mutated metastatic CRC who had progressed after one to two prior lines of therapy found that there was a median OS benefit to triplet therapy with encorafenib, binimetinib and cetuximab compared to investigator’s choice chemotherapy plus cetuximab. Median OS in the triplet-therapy group was 9.0 months (95% CI 8–11.4) versus 5.4 months (95% CI 4.8–6.6) in the control group, and ORR was 26% versus 2%, respectively. Median OS in the doublet-therapy group which received encorafenib and cetuximab was 8.4 months (95% CI 7.5–11.0) [41]. Grade ≥3 AEs in the triplet-therapy, doublet-therapy and control groups were 58%, 50% and 61%, respectively [41]. An updated analysis with an additional six months of follow-up was presented at the ASCO Gastrointestinal Cancers Symposium in January 2020. Patients in the triplet-therapy and doublet-therapy groups both had median OS of 9.3 months and ORR rates were similar: 27% (95% CI 21–33) in the triplet-therapy group versus 20% (95% CI 15–25) in the doublet-therapy group. These were superior to the control group which had a median OS of 5.9 months and ORR 2% (95% CI < 1–5%). Patients in the doublet-therapy group also had far fewer AEs, with no more than 6% of patients experiencing a Grade ≥3 AE. These findings led to FDA approval in April 2020 of encorafenib in combination with cetuximab for mCRC with disease progression after one or two prior regimens [42,43]. In the first-line setting, there is an ongoing Phase II trial (ANCHOR-CRC, NCT03693170) of triplet-therapy in patients with previously untreated BRAF V600E mutant mCRC [44]

BRAF mutations are most common in iCCA, as compared to extrahepatic CCA or gallbladder cancer. Mutations are present between 1% and 22% of cancers, but this incidence might be underestimated due to use of IHC studies rather than PCR [45]. Single-agent vemurafenib, a BRAF inhibitor, was evaluated in multiple cancers including CCA. It was associated with a 12% response rate (1 out of 8 PR) [46]. The ROAR study is a basket trial with a total of nine cohorts, involving 178 patients harboring rare cancers with BRAF V600E mutations. Patients were given combination dabrafenib and trametinib. The biliary cancer cohort included 35 patients, 80% of whom had been treated with two or more previous lines of therapy. Preliminary results presented at the ASCO 2019 GI symposium showed a response rate of 42% and a median overall survival of 11.7 months [47]. Additional data from this trial and other prospective trials are ongoing (NCT01713972, NCT01902173) [48].

## 6. NTRK

Neurotrophic tropomyosin receptor kinases are a family of proteins that includes TRKA (encoded by NTRK1), TRKB (encoded by NTRK2), and TRKC (encoded by NTRK3). They consist of an extracellular ligand-binding domain, transmembrane region, and an intracellular kinase domain. These tyrosine kinases are integral in neural development [49]. Under normal conditions, ligand binding leads to activation of the kinase domain, which leads to downstream signaling activation. These pathways include Ras–Raf–MAPK, PI3K–Akt–mTOR and PLCc–PKC [50]. Chromosomal rearrangement of the NTRK genes leading to gene fusion occurs in 1% of solid tumors across a wide variety of tumor types [51]. The resulting TRK fusion protein leads to cell transformation, growth, and proliferation.

Larotrectinib was recently granted FDA approval for TRK fusion cancer. The approval comes after a review of three single arm trials: LOXO-TRK-14001, NAVIGATE, and SCOUT trials. Fifty-five patients were enrolled in these trials ranging from 4 months to 76 years old; 71% of patients enrolled had continued response at 1 year; 55% also remained with progression free survival. In the studies, two patients had CCA; of these patients, one had a great response and the other patient had progression. Other GI cancers included in the study were four cases of colon cancer and one case of pancreatic cancer; all had a substantial change of greater than 30% in tumor size during treatment [52].

## 7. PARP

Poly (adenosine diphosphate–ribose) polymerases (PARP) are an 18-member family of enzymes that play an integral role in maintaining genome stability [53]. PARP-1 and PARP-2 are best known for their management of DNA damage repair (DDR) and are structurally similar [54]. Genomic instability is a common finding in human cancers; however, double strand breaks of DNA are considered the most severe type of DNA damage. These are mainly repaired via non-homologous end-joining (NHEJ) and by homologous recombination repair (HRR), which are complementary processes of DDR. When NHEJ is defective, PARP1-dependent end joining is activated to assist in the repair. In cancer cells, these PARP enzymes continue to repair DNA damage. PARP inhibitors act by preventing DNA repair by PARP enzymes in cancer cells and are particularly effective in HRR deficient cancer cells, resulting in cell death through a process known as synthetic lethality [55]. When BRCA genes, which are some of the genes that encode for proteins involved in HRR, are mutated, this allows for PARP inhibition via trapping of PARP at sites of single-strand DNA breaks, thus preventing repair of these strands and subsequently generating double strand DNA breaks. Germline BRCA mutations are found in 4–7% of pancreatic ductal adenocarcinoma ( PDAC) patients [56].

Kaufman et al. first evaluated the use of olaparib as monotherapy in a phase II multicenter non-randomized trial involving cases of advanced cancer and germline BRCA1/2 mutation [57]. Olaparib is known to have one of the higher potencies for PARP trapping [58]. Within the trial, 23 patients had PDAC and 74% of these patients had a BRCA2 mutation. This subgroup had a 21.7% response rate with CR in one patient and PR in four patients. Stable disease was seen in 35% of patients at week 8. The landmark POLO trial was a Phase III double-blind, placebo-controlled, multi-center trial randomized 154 patients with germline BRCA-mutated (gBRCAm) metastatic PDAC in a 3:2 fashion to olaparib or placebo until disease progression or unacceptable toxicity. Eligible patients had to have achieved at least four months of stable disease with front-line platinum-based chemotherapy. Median PFS was 7.4 months in the olaparib arm versus 3.8 months in the placebo arm. The median OS was not statistically significant at 18.9 months for olaparib versus 18.1 months for placebo, though this group may have had inherently better prognosis as an 18-month median survival for metastatic PDAC is longer than is typically seen in this population. Overall response rate among patients with measurable disease at trial initiation was 23% and 12%, respectively. A notable limitation of this study was the use of placebo as the control arm because discontinuation of all treatment after 4 months of chemotherapy is not the standard of care for PDAC when patients are still receiving benefits from therapy. Despite these limitations, the data from this trial led to the FDA approval of olaparib on 27 December 2019 for the maintenance treatment of adult patients with deleterious or suspected deleterious gBRCAm metastatic pancreatic adenocarcinoma whose disease has not progressed on at least 16 weeks of a first-line platinum-based chemotherapy regimen [56].

## 8. IDH

Isocitrate dehydrogenase (IDH) is an enzyme of the Krebs cycle that converts isocitrate to alpha-ketoglutarate (AKG). AKG is used as a cofactor in many enzymes such as DNA and histone modifiers. IDH mutations (IDH1 and IDH2) have mutational frequency of 15–20% in intrahepatic CCA (iCCA) [59]. Most mutations lead to a gain in function [60]. Mutated IDH triggers conversion of AKG to 2-OH-glutarate (2-HG). 2-HG then acts to competitively bind and inhibits the enzymes that use AKG as a cofactor, leading to gene dysregulation. Of note, IDH1 genetic aberrations do not carry any prognostic significance [61].

Ivosidenib is a potent first class IDH oral inhibitor of the mIDH1 enzyme. Seventy-three patients with known IDH-mutated CCA were treated with ivosidenib in a phase I-II trial. The trial achieved a median PFS of 3.8 months with a 6-month PFS of 40.1% and a 12-month PFS of 21.8% [62]. This led to the development of the ClarIDHy trial.

The ClarIDHy trial is a phase III randomized trial that includes pretreated CCA patients who received either ivosidenib or placebo. Results were presented as an abstract at ESMO 2019. Patients with advanced CCA and IDH1 mutations (185 in total) were assigned in a 2:1 ratio of ivosidenib to placebo. Interim results showed a PFS of 2.7 months versus 1.4 months. Six-month and 12-month PFS was 32% and 21.9%, respectively. However, overall disease control rate was 53% (stable disease and partial response) [63]. Final results of this trial are still pending. An additional trial evaluating another IDH1 inhibitor, BAY143602, is ongoing (NCT02746081).

## 9. FGFR-2

Fibroblast growth factor receptor (FGFR) fusions are present in 13–17% of CCA, but predominantly in iCCA. These fusions have been associated with a more favorable prognosis.

A recent phase II study evaluated BGJ398 in patients with an FGFR2 fusion, mutation, or amplifications who were previously treated with chemotherapy. In the study, all patients with FGFR2 fusions had radiological responses (48 patients out of 61 total). Overall response rate was 18.8% in fusion patients with a disease control rate of 83.3%. Median progression free survival was 5.8 months [64].

Data from the FIGHT-202 study was presented at ESMO 2019. This phase II study evaluated pemigatinib, a selective, potent, oral FGFR 1, 2, and 3 inhibitor, in patients with previously treated, locally advanced metastatic CCA (NCT02924376). The study accrued 146 patients who were assigned to cohorts A (FGFR2 gene rearrangements/fusions), B (other FGF/FGFR gene alterations), or C (no FGF/FGFR gene alterations). There were 107, 20, and 18 patients in each cohort, respectively. The overall response rate was 35.5% with three patients achieving complete responses. Median duration of response was 7.5 months with a disease control rate of 82%; median progression free survival was 6.9 months and median overall survival was 21.1 months. Overall survival data were not yet mature at the time of reporting at ESMO 2019 [65].

## 10. Conclusions

In summary, there are increasingly more targetable molecular mutations in GI malignancies with practice-changing implications (Table 1, Table 2, Table 3 and Table 4). We have highlighted nine major molecular profiles that have been or are becoming practice-changing in GI oncology. We expanded on known molecular markers and discussed potential new targets on the horizon. dMMR/MSI-H predicts benefits from immunotherapy in any solid tumor. Extended RAS and BRAF V600 mutations in mCRC confer benefits from EGFR pathway inhibition. While combination BRAF and MEK inhibition is proven to be of benefit in V600 mutated cholangiocarcinoma, additional EGFR inhibition is required in V600 mutated colorectal carcinoma. HER-2 amplification guides therapy with trastuzumab and other HER2-directed combination therapies in gastric, cholangiocarcinoma and colorectal carcinoma. PD-L1 CPS scores in refractory metastatic GEJ cancers predict benefits with pembrolizumab. Newer approaches are the maintenance of PARP inhibition in patients with germline BRCA mutation in pancreatic adenocarcinoma, and IDH and FGFR inhibition in biliary tract cancer. A comprehensive understanding of the mechanisms of the prevalent genetic aberrations is quickly becoming paramount in prognostication and tumor-based therapeutic regimens. When performed and interpreted appropriately, biomarker selection and targeted therapies produce dramatic improvements in disease control and survival as well as a more tolerable side effect profile as compared to traditional therapies.

## Figures and Tables

**Table 1 cancers-12-01168-t001:** Methods of molecular testing for genetic mutations mentioned and relevant malignancies. Poly (adenosine diphosphate–ribose) polymerases (PARP), isocitrate dehydrogenase (IDH), and BRAF mutations can be detected by next generation sequencing (NGS) alone, whereas deficient mismatch repair (dMMR)/microsatellite instability (MSI), NTRK, FGFR2, and HER2 require additional testing for confirmation of positivity. PD-L1 combined positive score (CPS) can be calculated from immunohistochemistry (IHC) alone.

Molecular Mutation	Methods of Testing	Relevant GI Malignancies
dMMR/MSI	NGS *plus* IHC *or* PCR	All solid tumors
PD-L1 CPS	IHC	GEJ, gastric, esophageal cancer
NTRK Fusion	NGS *plus* FISH	All solid tumors
PARP Mutations	NGS	Pancreas
IDH Mutations	NGS	CCA
FGFR2 Fusion	NGS *plus* FISH	CCA
HER2 Amplification	NGS *plus* IHC *or* FISH	GEJ, gastric, CRC, gallbladder
BRAF Mutations	NGS	CRC and CCA

IHC = Immunohistochemistry. NGS = Next Generation Sequencing. FISH = Fluorescent In Situ Hybridization. PCR = Polymerase Chain Reaction.

**Table 2 cancers-12-01168-t002:** Microsatellite instability-high (MSI-H) status and therapeutic targets.

Study	Phase	Tumor Site	Size (n)	Arm	ORR	PFS	OS
Tabernero et al. KEYNOTE-062 [12]	III	Advanced gastric and GEJ adenocarcinoma	763	Pembrolizumab	25% (CPS ≥ 10 group)	2.9 months (CPS ≥ 10 group)	Median 17.4 months (CPS ≥ 10 group)
					14.5% (CPS ≥ 1)	10.6 months (CPS ≥ 1)	Median 10.6 months (CPS ≥ 1)
					57.1% (MSI-H, CPS ≥ 1)	11.2 months (MSI-H, CPS ≥ 1)	Not reached (MSI-H, CPS ≥ 1)
				Pembrolizumab + Chemotherapy	64.7% (MSI-H, CPS ≥ 1)	Not reached (MSI-H, CPS ≥ 1)	Not reached (MSI-H, CPS ≥ 1)
Le et al. [7]	II	Colorectal adenocarcinoma and non-colorectal adenocarcinomas	41	Pembrolizumab	40% (dMMR CRC)	78% (dMMR)	Not reached (dMMR)
					71% (dMMR non-CRC)	67% (dMMR non-CRC)	5.45 months (dMMR non-CRC)
					0% (pMMR CRC)	11% (pMMR CRC)	2.2 months (pMMR CRC)
Overman et al. CheckMate-142 [9]	II	Metastatic colorectal adenocarcinoma	45	Nivolumab + Ipilimumab	64%	Not reached	Not reached
Chalabi et al. NICHE Study [10]	II	Early-stage colorectal adenocarcinoma	40 total; 35 pts were evaluable	Nivolumab + ipilimumab	100% (dMMR group)		
					27% (pMMR group)		

CPS = combined positive score; dMMR = deficient mismatch repair; pMMR = proficient mismatch repair; CRC = colorectal cancer; pts = patients.

**Table 3 cancers-12-01168-t003:** HER2 status and therapeutic targets (selected studies).

Study	Phase	Tumor Site	Sample Size (n)	Arm	ORR	PFS	OS
Bang et al. ToGA [25]	III	Advanced gastric or GEJ cancer	594	Trastuzumab + chemotherapy	47%	Median 6.7 months	Median 13.8 months
Hainsworth et al. MyPathway [30]	IIa	Advanced solid tumors	230 total; 37 mCRC cancers; 11 biliary cancers	Trastuzumab + pertuzumab	38% (mCRC)		
					37.5% (biliary cancer)		
Sartore-Bianchi et al. HERACLES [31]	II	Metastatic CRC	27	Trastuzumab + lapatinib	30%	Median 4.8 months	Median 10.6 months

GEJ = gastroesophageal junction; mCRC = metastatic colorectal cancer.

**Table 4 cancers-12-01168-t004:** PD-1/PD-L1 status and therapeutic targets (selected studies).

Study	Phase	Tumor Site	Sample Size (n)	Arm	ORR	PFS	OS
Fuchs et al. KEYNOTE-059 [14]	II	Advanced gastric or GEJ adenocarcinoma	259	Pembrolizumab	11.6% (overall)	Median 2.0 months (overall)	Median 5.6 months (overall)
					15.5% (PD-L1 positive)		Median 5.8 months (PD-L1 positive)
Kang et al. ATTRACTION-2 [15]	III	Unresectable advanced or recurrent gastric or GEJ adenocarcinoma	493	Nivolumab	11%	Median 1.61 months (overall)	Median 5.26 months (overall)
							Median 5.22 months (PD-L1 positive)
Shah et al. KEYNOTE-180 [66]	II	Advanced, metastatic esophageal cancer	121	Pembrolizumab	9.9% (overall)	Median 2.0 months (overall)	Median 5.8 months (overall)
					14.3% (SCC)	Median 2.1 months (SCC)	Median 6.8 months (SCC)
					5.2% (ADC)	Median 1.9 months (ADC)	Median 3.9 months (ADC)
					13.8% (PD-L1 positive)	Median 2.0 months (PD-L1 positive)	Median 6.3 months (PD-L1 positive)
Kojima et al. KEYNOTE-181 [22]	III	Advanced or metastatic esophageal carcinoma	628	Pembrolizumab	21.5% (overall)	Median 2.6 months (CPS ≥ 10)	Median 9.3 months (with CPS≥ 10)
					22% (SCC)		Median 10.3 months (SCC with CPS ≥ 10)
					18% (ACC)		Median 6.3 months (ADC with CPS ≥ 10)

GEJ = gastroesophageal junction; PD-L1 = programmed death-ligand 1; SCC = squamous cell carcinoma; ADC = adenocarcinoma; CPS = combined positive score.

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
