# Peer review of "Recent Advances in Targeted Therapies for Advanced Gastrointestinal Malignancies"

_cancers, 2020, doi:10.3390/cancers12051168_

Round 1

Reviewer 1 Report

To the authors, 

The authors have provided a very generalized overview of the literature regarding GI and hepatobiliary malignancies.  However, the paper could be further refined. 

1.  The title is rather odd.  Hepatobiliary is also c/w with GI, why is this differentiated in the title? It is not separated in the body of the manuscript. 

2.  I would REQUIRE that tables or Figures be considered to reduce the monotony of the manuscript. 

3.  I would add the new Chalabi paper re: Neoadjuvant therapy.

4.  The order for discussing the molecular markers seems rather random. For instance, IDH and FGFR-2 are listed before HER2 which is more applicable in gastric and colorectal CA. BRAF is listed last?  

5.  Add the FDA approval of the doublet for BRAF. 

6.  I would mention the Anchor study. 

7.   I would revised the manuscript and try to find a way to reduce the monotony and not make it so bland.  

Author Response

  1. The title of the manuscript has been revised to exclude hepatobiliary and now reads “Recent advances in targeted therapies for advanced gastrointestinal malignancies.”
  2. Tables have been added to summarize selected studies in the MSI High, PD-L1 and HER2 sections.
  3. The Chalabi et al. paper (line 74) was added to the manuscript under the MSI High section to detail the use of neoadjuvant ipilimumab and nivolumab prior to surgery in patients with early-stage CRC.
  4. The order of the manuscript has been revised to the following order: MSI High, PD-L1, HER2, BRAF, NTRK, PARP, IDH and FGFR2.
  5. The recent FDA approval of the doublet encorafenib/cetuximab was added to the BRAF section (line 205).
  6. Discussion of the ongoing Phase II ANCHOR-CRC study for BRAF-mutant metastatic CRC was added to the BRAF section (line 207).
  7. The manuscript was edited to make it more concise and tables were added (see Tables 1-4, attached document).

Reviewer 2 Report

This is  a major review on the targeted and immunotherapy of GI cancers. This aim seems to me too ambitious which lead to several significant misstakes. The major problem is that it is not comprehensive: anti-EGFR therapy is completely missing although it is the basis of the treatment of CRC. Second, anti-VEGF/VEGFR therapy is also missing although it is now the other major arm of CRC therapy but also some other GI cancers as well. It is also a problem that the authors do not comment contrasting/contradictory data or protocols.

Point-by point

L44 2. MSI. There are no % data on the MMRD incidence in GI cancers in details. That could help to understand why you have different RRs in various GI cancers for immuncheckpoint inhibitors....There is no comment on the rationale of using Pembro in gastric cancer without determining MMRD as compared to CRC, what is the connection of PDL1 expression and MSI status in GI cancers etc. It is also missing some comment on the incidence of EBV in gastric cancer and a potential link between ICI efficacy and viral infection...I would expect also a comment on HP positive/negative gastric cancers concerning ICI therapy efficacy....

L84 3. PDL1. the title missleading: PD1/PDL1 is correct. The inhibitors must be indentified by their appropriate target. Why not to start with esopgaeal SCC followed by gastric, than CRC. There is no comment on Keynote 181, where SSC and ADC of esophageal cancers were mixed together!!!!!!

L153 PARP. This chapter is on PDAC only. L167: PDAC is not hepatobiliary cancer . There are no % data on incidence of HR and or BRCA mutations in various GI cancers

L220 FGFR(2) is missing. There is confusion if only fusion or other gene defects are important: mutation, amplification (L222), incidence rate is necessary in %.

8. HER2 must be disscussed much earlier. L237 it must be stated clearly that we are talking about adenocarcinoma in case of esophagel cancer!!!!There is a total confusion concerning overexpression (protein) and amolification of HER2. In each trial, one would expect to clearly kvote the correct indication statement!!!!L252 even state HER2 mutated (amplified only). L257: not EGFR but EGFR1/2 inhibitors.L260,262 Her2 overexpresison term used but I think it is amplification in each case. 

9. BRAF. it is not cited that now BRAF mutation is a strong negative predictor of anti-EGFR antibody therapy of CRC similarly to RAS mutation. 

Author Response

  1. Anti-EGFR therapy was intentionally not included in this discussion as our aim was to review recent advances in targeted therapies for GI malignancies, and anti-EGFR therapy has already been well-established.
  2. MSI section: The prevalence of dMMR in GI malignancies has been added: 13% for Stage 1-2 cancers, and 10% in Stage 3-4 cancers (line 45). Colorectal cancer (CRC) has the highest prevalence of dMMR tumors, ranging from 5-20% (line 47). Discussion about EBV in gastric cancer and the excellent response rate with checkpoint inhibitor therapy in Kim et al’s study has also been added (line 120). HER2 positivity and the influence of checkpoint inhibition is discussed (line 128).
  3. PD-L1: The title of this section has been revised to “PD-1/PD-L1” (Line 100). There was not an extensive discussion of the analysis of KEYNOTE-181 as we aimed to focus on the overall findings of the study, not provide a critical review.
  4. PARP: This section was revised to remove the comment about PDAC being a hepatobiliary cancer. Data regarding germline BRCA mutations, found in 4-7% of pancreatic adenocarcinoma patients, was added (line 269).
  5. FGFR2: The incidence rate of FGFR fusions is listed as 13-17% (line 314) and it is stated that fusions are associated with a more favorable prognosis.
  6. HER2: This section was moved to the fourth section of the manuscript (line 154). Descriptions have been revised to state that in the MyPathway study, patients had HER2 amplification/overexpression (line 177). HER2 positivity, as used in the HERACLES trial, was also defined (line 183). The mention of EGFR with regards to the HERACLES trial was revised to “EGFR 1/2 inhibitors” (line 186).
  7. BRAF: The topic of BRAF mutations and lack of response to EGFR inhibition was further elaborated upon. A reference to Gong et al’s comprehensive review paper regarding the lack of response to EGFR inhibitors in tumors with extended RAS mutations, BRAF mutations and right-sided colon tumors was added (line 209). A specific EGFR inhibition section was intentionally excluded as our goal was to review recent advances in targeted therapies (see above, point #1).

Reviewer 3 Report

The manuscript looks good and well-written.

However, I think that the review could me more comprehensive. For example, they described FGFR2-targeted therpy in CCA. Then, they can also describe anti-FGFR2 therapy in FGR2-amplified gastric cancer, althogh teh attempt is not so successful at present (Catenacci et al. JCO 2020). In regard to HER2, they can describe the promsing activity of Daiichi-Sankyo's ADC in HER2-positive GC.

I also think the manuscript is not so well-organized. For example, IDH, FGFR2, and HER2 are molecular targets, whereas MSI-H and PD-L1 are biomarkers for anti-PD-1 antibodies. Anti-EGFR antibodies for CRC were described in the section of BRAF.

The manuscript needs more editing and more recent references. I attached a file for these.

Author Response

  1. Although anti-FGFR2 therapy has been studied in FGR2-amplified gastric cancer, this has not been successful and our intention with this manuscript is to describe therapies that have been successful.
  2. Description of trastuzumab deruxtecan for gastric cancer has been added in the HER2 section (line 165). Shitara et al’s Phase I study and the Phase II DESTINY-Gastric-01 trial were discussed. (line 168 and line 172, respectively).
  3. The manuscript has been reorganized to move the more common biomarkers (MSI high, PD-1/PD-L1 and HER2) toward the beginning and the less frequent molecular targets toward the end.

Round 2

Reviewer 3 Report

The manuscript looks good.

I have one edit and one comment.

  • Page 2, line 49. 'trials have exploring' should be read as 'trials exploring'.
  • Page 7, line 322. Pemigatinib was approved by the FDA, several days ago.